# Archaea Microbiome Dysregulated Genes and Pathways as Molecular Targets for Lung Adenocarcinoma and Squamous Cell Carcinoma

**DOI:** 10.3390/ijms231911566

**Published:** 2022-09-30

**Authors:** Matthew Uzelac, Yuxiang Li, Jaideep Chakladar, Wei Tse Li, Weg M. Ongkeko

**Affiliations:** 1Division of Otolaryngology-Head and Neck Surgery, Department of Surgery, University of California, San Diego, CA 92093, USA; 2Research Service, VA San Diego Healthcare System, San Diego, CA 92161, USA

**Keywords:** LUAD: lung adenocarcinoma, LUSC lung squamous cell carcinoma, ROS: reactive oxygen species, TCGA: the Cancer Genome Atlas, GSEA: Gene Set Enrichment Analysis, NES: normalized enrichment score, SHAP: SHapley Additive exPlanations, RFE: recursive feature elimination

## Abstract

The human microbiome is a vast collection of microbial species that exist throughout the human body and regulate various bodily functions and phenomena. Of the microbial species that exist in the human microbiome, those within the archaea domain have not been characterized to the extent of those in more common domains, despite their potential for unique metabolic interaction with host cells. Research has correlated tumoral presence of *bacterial* microbial species to the development and progression of lung cancer; however, the impacts and influences of archaea in the microbiome remain heavily unexplored. Within the United States lung cancer remains highly fatal, responsible for over 100,000 deaths every year with a 5-year survival rate of roughly 22.9%. This project attempts to investigate specific archaeal species’ correlation to lung adenocarcinoma (LUAD) and lung squamous cell carcinoma (LUSC) incidence, patient staging, death rates across individuals of varying ages, races, genders, and smoking-statuses, and potential molecular targets associated with archaea microbiome. Archaeal species abundance was assessed across lung tissue samples of 527 LUAD patients, 479 LUSC patients, and 99 healthy individuals. Nine archaeal species were found to be of significantly altered abundance in cancerous samples as compared to normal counterparts, 6 of which are common to both LUAD and LUSC subgroups. Several of these species are of the taxonomic class Thermoprotei or the phylum Euryarchaeota, both known to contain metabolic processes distinct from most bacterial species. Host-microbe metabolic interactions may be responsible for the observed correlation of these species’ abundance with cancer incidence. Significant microbes were correlated to patient gene expression to reveal genes of altered abundance with respect to high and low archaeal presence. With these genes, cellular oncogenic signaling pathways were analyzed for enrichment across cancer and normal samples. In comparing gene expression between LUAD and adjacent normal samples, 2 gene sets were found to be significantly enriched in cancers. In LUSC comparison, 6 sets were significantly enriched in cancer, and 34 were enriched in normals. Microbial counts across healthy and cancerous patients were then used to develop a machine-learning based predictive algorithm, capable of distinguishing lung cancer patients from healthy normal with 99% accuracy.

## 1. Introduction

Over recent decades, the human microbiome has undergone increased investigation. Studies estimate that roughly 500–1000 bacterial species are present within the human body at any time, with their sheer quantity significantly outnumbering human cells [1]. With such abundance observed, more research is being conducted regarding the human microbiome’s correlation to individual health, including cancer development and progression [2]. Though the gut microbiome’s influence on human carcinogenesis is widely accepted, mechanisms of influence are poorly understood. Studies show bodily circulation of gastrointestinal microbial metabolites to be a plausible means of carcinogenesis [2]. Alternatively, evidence continues to emerge suggesting the correlation of *tumoral* microbial composition with cancer incidence and prognosis [3]. In the occurrence of microbes migrating to pre-tumoral sites, species may be more capable of direct chemical influence [2]; specifically, popular mechanisms describe these species’ metabolites’ effects on bodily epithelial tissue [4]. Microbial metabolic generation of reactive oxygen species (ROS), for example, is known to cause oxidative cellular injury, ultimately promoting cancer development [5,6]. Among other common metabolites, ROS are highly reactive compounds shown to chemically degrade vital cellular proteins and macromolecules. As produced by many microbial species, ROS induced dysregulation of cell cycle or inflammatory proteins and genes, for instance, cause contribute to carcinogenesis [5,6]. Butyrate and hydrogen sulfide are also among other microbial metabolites commonly correlated to cancer development [7]. In addition to cell cycle regulation, these metabolites are known to regulate host immune response, modulating immune cell activity and encouraging inflammatory induced carcinogenesis [8,9].

Given the human microbiome’s extensive correlation to carcinogenesis, studies now attempt to characterize specific microbial landscapes of which individuals are at greater risk; a recent study developed an algorithm for highly sensitive cancer diagnosis using the microbial compositions of 18,116 human blood samples across 33 cancer types [10]. The ability to accurately diagnose an array of cancers using a sole blood sample would greatly advance cancer treatment and health services. Equally provocative would be the potential of these archaeal species, their metabolites, or gene dysregulations they cause, to be molecular targets for cancer therapy. Many studies choose to investigate the human *bacterial* microbiome, however, often overlooking the archaeal domain altogether. Though less abundant within the human body, archaea are also thought to correlate significantly to individual health and disease development [11]. Through the course of evolution, archaeal species have developed rather extreme survival dependencies and metabolic properties [12]. Thus, the human archaeal microbiome should differ significantly from the bacterial microbiome with regard to health and disease influence; unique metabolic effects might influence tumoral growth to a greater extent than bacterial counterparts. In fact, several archaeal phylums and classes are known to exist in the human body: notably, methanogens, Desulfurocaccales, Sulfolobales, Thermoproteales, Nitrosospharerales, and Halobacteriales [13]. Comparatively, few studies have analyzed the relationship between human archaea populations and individual health as a whole.

Lung cancer in particular has seen extensive correlation to the tumoral microbiome, exhibiting the highest morbidity rates and the second-highest diagnosis rates of all cancer type [14,15]. Lung adenocarcinoma (LUAD) is the most prevalent form of lung cancer within the United States, accounting for approximately 40% of all lung cancer cases [16]. Tumoral sites of this cancer are often in the lung periphery, where many microbial species are also present [17]. Studies have shown a variety of archaeal species to be present within lung tissue [13]. Lung squamous cell carcinoma (LUSC) trails LUAD in prevalence rates and is found in central lung and airway sites [18]. Given lung cancer’s considerable severity across the nation, as well as the known presence of archaeal species within these tumoral sites, LUAD and LUSC lung tissue samples were chosen for analysis throughout this experiment. This study aims to investigate the potential relationship between a patient’s archaeal lung microbiome with the development and progression of LUAD and LUSC.

527 intratumoral lung tissue samples were analyzed among LUAD patients, and 479 samples were analyzed among LUSC patients. A total of 99 normal adjacent lung samples were used as a means of assessing cancer development correlations. Samples were sourced from The Cancer Genome Atlas (TCGA) online database (accessed on 1 August 2021). Within cancer samples, seven variables of interest were analyzed: patient age of diagnosis, sex, race, smoking history, pathologic cancer stage, neoplasm development, and effectiveness in received therapy treatments. Patient survival length after diagnosis was also analyzed, again in relation to samples’ archaeal compositions.

This study aims to identify potential correlation of tumoral tissue microbiome composition with incidence and progression rates of LUAD and LUSC. In characterizing species-level discrepancies, a cancer’s course of progression can be better predicted. Analysis of corresponding archaeal metabolites and dysregulated cellular signaling pathways will provide further insight into potential causational mechanisms of carcinogenesis. This study initially compares the archaeal microbiome of healthy normal samples with that of both LUAD and LUSC samples. Individual microbial species of significantly different abundance across cancer and non-cancer samples were next identified and investigated. Microbial species of differential abundance *within* cancer subgroups were also identified; this served to identify species of potential significance regarding worsened cancer progression. Several oncogenic and metabolomic gene sets were analyzed for dysregulation in association to altered abundance of significant archaeal species. Metabolic dysregulation of these cellular pathways might act as a precise mechanism for archaeal species’ influence on oncogenesis. Lastly, a machine learning-based predictive algorithm was trained to distinguish healthy individuals from cancer patients based solely on the microbial landscape of lung tissue samples.

## 2. Results

### 2.1. Microbial Contamination Correction

After download and extraction of raw microbial counts, species abundance data was further processed to remove any potential contaminant species from future analyses. The Biospecimen Core Resource (BCR) employs standard operation procedures to attain molecular analyte data of TCGA samples; in processing tissue, these procedures might introduce microbes not originally present within samples. Species that display no variation in abundance along with changes in total microbial abundance (across many samples) are supposed to have been introduced after sample extraction. Hence, species that exhibit this behavior were removed from further assessment, as they are likely contaminants.

Spearmen’s correlations were used to identify these contaminants. Individual Spearmen’s rank coefficients were calculated for the 424 microbes common to both LUAD and normal adjacent samples, each relating abundance of a particular species to abundance of all microbes within that sample. Coefficients were similarly calculated for the 447 species common to LUSC and normal adjacent samples. Species of insignificant correlation between these two variables were removed using the generated coefficients. This allows for increased confidence in deeming any species significant in future analyses. Among LUAD and normal samples, 294 species were regarded contaminants, leaving 130 species to be analyzed throughout the remainder of the study. A total of 334 contaminant species were found across LUSC and normal samples, leaving 113 species for analysis. (Appendix A).

### 2.2. Lung Tissue Microbial Landscaping

Dfferences in lung microbiome composition between healthy samples and cancer samples were initially identified. Individual comparison of each species’ abundance across each sample of each cancer status would prove rather a convoluted analysis. Additionally, visualization of any results would be difficult. As such, Multidimensional Scaling (MDS) ordination was used to assess the extent of similarity in microbial reads across patients. This means of ordination considers the abundance of each species within samples, reducing each patient’s microbial landscape into two arbitrary dimensions. Using this method, patients’ wholistic landscapes could be compared against one another, grouped by cancer status, in a more efficient manner.

The 527 LUAD samples were reduced and plotted alongside 99 normal adjacent samples. A total of 479 LUSC samples were similarly plotted against the same 99 adjacent lung samples. (Figure 1). The respective MDS plots depict samples’ lung microbiome compositions at a relative scale, with each data point representing a patient. Points in close proximity to one another represent samples of similar abundance values across many species, while points of greater distance correspond to samples with greater variation in abundance. It can be noted that LUAD samples occupy a rather distinct region of the ordination plot, as compared to the region of adjacent samples. Similarly, the locations of LUSC samplessuggest that patients of this cancer also contain a microbial environment unique from that of normal patients. Hence, it can be understood that among samples, patients diagnosed with LUAD or LUSC contain considerably different lung microbial compositions than normal counterparts. It should be noted the distinct “horseshoe”-like shape of both ordination plots. Commonly referred to as the “horseshoe effect”, this occurrence is observed in ordination analyses, and often signifies poor measurability of sample dissimilarity. In the case of microbial analyses, however, the horseshoe effect signifies greater *distinction* among microbial communities [19]; as observed, the horseshoe effect implies a lack of comparability in microbiome composition between cancer and normal samples, largely due to the immensity of abundance differences. Such distinction may prove of great use in discerning cancer samples from normal counterparts. The presence of this ordination pattern should not suggest incompatibility among samples, but should rather serve as a means of confirmation that these microbial landscapes truly differ between cancerous and normal tissue.

Indeed, cancer and normal samples showed minimal difference in total β-diversity. β-diversity characterizes differences in the extent of total microbial variation across samples. β-diversity was calculated across all LUAD samples and plotted against the calculated β-diversity of normal samples (Figure 1). β-diversity of LUSC patients was also plotted against that of the same normal patients (Figure 1). This suggests that despite the considerable differences noted in microbial composition across cancer statuses, the sheer counts of species present across samples are relatively comparable. This is expected, as β-diversity of a tumoral-microbiome does not differ considerably from that of a non-malignant tissue. This serves to confirm the significance of landscape variation as identified through the above MDS ordination; total diversity is comparable between cancer and normal samples, suggesting that the observed diversity across specific species is attributable to true variation, and not to other factors such as differences in microbial profiling.

### 2.3. LUAD and LUSC Microbiome Clinical Significance

With the microbial environment of cancerous lung tissue samples differing considerably from that of normal samples, the study next attempted to characterize *specific* differences in composition, as well as exact significances in variation. Abundance of individual species were compared across LUAD and normal samples, as well as LUSC and normal samples. In this way, variation in abundance of specific species could be identified. Kruskal–Wallis testing was employed in order to compared abundance reads across samples. This analysis determines the significance of differences in each species’ abundance across all samples of a cancer status. A threshold value of *p* < 0.05 was used to distinguish microbes of significantly altered abundance in samples of a particular status as compared to others.

Of the 130 microbes common to both LUAD and normal patients, 8 were found to be of considerably different abundance (Table 1). Box plots were generated for each of these species depicting their observed abundance value within each sample, grouped by cancer status. A phylogram was constructed to visualize the relationship of these 8 microbes, with the corresponding abundance box plots of *Methanosarcina Mazei Go1* and *Methanosarcina Barkeri str. Fusaro* included alongside (Figure 2). The differences in microbial composition as identified through the above MDS ordination can largely be attributed to these 8 species, as they were found to differ to the greatest extent between cancerous and normal tissue samples.

Similarly, of the 113 species common to both LUSC and normal samples, 7 were of significantly altered abundance across groups (Table 1). These microbes were also related to one another within a phylogram, with the abundance box plot of *Methanosarcina Barkeri str. Fusaro* again included (Figure 2). These 7 species largely account for microbial composition differences between LUSC and normal patients, as suggested through MDS ordination.

After identification of species differentially abundant across cancer and normal patients, samples were next analyzed to assess abundance significance *within* cancer groupings. These analyses would serve to further reveal any importance the lung microbiome might play in cancer progression across patients of differing race, age, or cancer stage for instance. Specifically, the variables assess across cancer samples included age of diagnosis, sex, race, smoking history, pathologic cancer stage, neoplasm presence, and outcome of any received therapy treatments. Kruskal–Wallis testing was again used to conduct this analysis, with species abundance compared individually across patients, grouped by the above variables. A threshold value of *p* < 0.05 was used to claim significance in relation.

With 130 microbes across LUAD samples and 7 variables of interest, 910 potential relationships were explored for significance. Of these, 49 were deemed significant, involving 37 unique species in total (Table 2A). Box plots displaying each sample’s abundance, grouped by variable, were then constructed for each significant microbe. A phylogram was created to display the relationship between these microbes, with abundance box plots shown for uncultured methanogenic archaeon, uncultured crenarchaeote, *Aeropyrum pernix*, and *Vulcanisaeta distributa DSM 14429* (Figure 3).

791 relationships were analyzed for significance of the 113 LUSC present species across 7 variables. A total of 20 of these were deemed significant, corresponding to 18 total microbial species (Table 2B). Similar box plots were produced as above, depicting a species’ abundance across patients of similar groupings by variable. The plots of *Methanothermococcus okinawensis IH1* and *Methanocaldococcus infernus ME* were included alongside a phylogram of significant microbes (Figure 3).

This study also attempted to identify microbial species significantly correlated to better or worse survival rates of patients. Cox Regression analysis was used to determine whether high or low abundance of a particular species generally related to increased or decreased survival times of LUAD and LUSC samples. This means of analysis defines the abundance of each species within a sample as a binary value: either “high” or “low” expression. As such, all samples of “high” abundance can be compared against those of “low” abundance. The time of which patients were last assessed, along with their vital status, was used to conduct said analyses regarding species’ relations to patient survival time. A threshold value of *p* < 0.05 was used to identify microbes of significant correlation to patient survival.

11 of the 130 species found across LUAD samples were found to be of significant relation to the length of which patients lived after diagnosis (Table 3). Survival plots were generated for significant species, each displaying the proportion of patients alive after a given length of time, grouped by “high” or “low” abundance of a particular species. A phylogram was constructed to visualize the relation of significant species, with corresponding survival plots included for *Thermoproteus* sp. *IC-062*, *Thermoproteus uzoniensis*, and *Pyrobaculum islandicum* (Figure 4).

Of the 113 LUSC present microbes, abundance readings of 4 species were found to significantly correspond to altered survival rates in patients (Table 3). Survival plots were again generated for significant species. The plot for uncultured Methanobacteriales archaeon was included alongside the respective phylogram (Figure 4).

### 2.4. Oncogenic and Metabolomic Signaling Pathway Dysregulation

The 8 microbes differentially abundant in LUAD patients and 7 differentially abundant in LUSC patients were further analyzed to reveal correlations to patient gene expression. Abundance values were simplified to a binary “high” or “low” presence based on their individual relation to the median abundance of their respective species.

Patient gene expression counts were downloaded from TCGA for LUAD, LUSC and adjacent normal samples. Read counts for 60,660 genes were aligned with the 527 LUAD, 479 LUSC, and 99 normal adjacent samples used above in microbial analysis. Kruskal–Wallis testing was used to correlate each gene’s expression value across patients to high or low microbe abundance for each of the significant taxa above. A total of 3554 genes were found to be differentially expressed in LUAD-normal analyses, and 3884 were differentially expressed in LUSC-normal analyses. These genes display considerable alteration in expression in correlation to varied abundance of the prior identified significant archaeal species. As such, they were next analyzed to determine potential enrichment effects on oncogenic signaling pathways, providing a more definitive relation to these species’ correlation with cancer incidence.

189 oncogenic gene sets were sourced from the molecular signatures database (MSigDB) (accessed on 8 July 2022). These sets contain common features of known oncogenic signaling pathways typically dysregulated in cancer cells. A total of 225 metabolomic gene sets were similarly obtained, each containing genes commonly associated in several cellular metabolic pathways. Gene Set Enrichment Analysis (GSEA) was performed in order to estimate the enrichment effects on these pathways of the 3554 genes significant to LUAD and normal samples. Likewise, this analysis was repeated for the 3884 genes significant to LUSC and normal samples.

Regarding LUAD, 2 oncogenic gene sets were positively enriched (nominal *p*-value < 0.05) in cancer samples as compared to normals, with respective normalized enrichment scores (NES) of 1.52 and 1.50. In LUSC groupings, 6 oncogenic gene sets were positively enriched in cancer samples, with NESs > 1.59. Genes in these pathways were of significantly greater expression in cancer samples than normals. Interestingly, 34 oncogenic gene sets were negatively enriched in LUSC samples, with NESs < −1.46: 7 of which displayed nominal *p*-values < 0.01. Genes in these pathways were of significantly *lower* expression in cancer samples than normals.

2 metabolomic gene sets were positively enriched among LUAD samples as compared to normals, with *p*-value < 0.05. In LUSC samples 3 metabolomic gene sets were significantly positively enriched, with *p*-value < 0.01. These genes were present to a greater extent in archaeal-dysregulated cancer samples, suggesting an increase in their metabolic capacity. Four metabolomic gene sets were significantly enriched in normal samples, with *p*-value < 0.01.

Enrichment plots were generated displaying a gene set’s enrichment score (ES) in relation to each of feature’s relative expression across samples (Figure 5).

To understand the extent of significance of correlation between these genes and pathways, NESs were plotted against each gene set’s nominal p-value and FDR q-value (Figure 6). In comparing LUSC and normal samples, 84 pathways displayed a q-value < 25%, and 40 displayed a *p*-value < 0.05. Between LUAD and normal samples, 2 displayed a *p*-value < 0.05.

### 2.5. Archaeal Microbiome-Based Machine Learning Algorithm for Clinical Diagnosis

Supervised machine learning (ML) algorithms were trained to distinguish lung cancer-diagnosed patients from healthy individuals based solely on lung tissue sampling. Separate algorithms were developed for LUAD and LUSC, each analyzing the archaeal abundance of key species across lung tumor tissue and normal adjacent tissue. Light gradient boost machine (LGBM) achieved 99% accuracy in diagnosis of LUAD patient test sets (ROCAUC = 1.000, PRAUC = 0.994), and 100% accuracy in LUSC patient test sets (ROCAUC = 1.000, PRAUC = 0.993) (Figure 7A). Models were interpreted with SHapley Additive exPlanations (SHAP) values. For both LUAD and LUSC subsets, *uncultured marine group II/III euryarchaeote KM3_87_G01* served as the primary predictive feature (Figure 7B); predictive features are specific archaeal species most useful to the algorithm in distinguishing cancer samples from normals. Recursive feature elimination (RFE) was used to select a collection of features that most effectively distinguish cancer patients from normal adjacents. A total of 20 significant features were resultant of the LUAD model, and 2 significant features were resultant of the LUSC model (Figure 7C).

## 3. Discussion

### 3.1. Phylogenic Classification

As outlined above, the archaeal lung microbiome seems to correlate heavily to both cancer presence, and cancer progression. Nine species were found to be of significantly altered abundance between cancer and normal samples, 6 of which were common to both LUAD and LUSC patients. A total of 48 species were differentially abundant within cancer samples of similar pathologic stage, therapy outcome, etc., with 8 common to both LUAD and LUSC patients. The following portion of this discussion investigates these microbial species and their characteristics. This section introduces known metabolic properties of these species and proposes plausible hypothetical mechanisms of microbial impact on host tissue. It should be noted that due to the lack of research regarding archaeal interaction with human tissue, some suggested means of influence have not been experimentally confirmed.

Microbial species are known to interact with proximal host cells through a variety of mechanisms [4]. Common to both bacterial *and* archaeal species, several metabolites have been proven capable of translocating into host cells and interfering with typical cellular function. Of these identified metabolites, reactive oxygen species (ROS), bile acids (BA), butyrate, hydrogen sulfide, and N-nitroso compounds are more commonly observed [7,20]. These metabolites are known to influence immune response, contribute to cellular inflammation, and induce tumorigeneisis [20].

*Methanosarcina barkeri str. Fusaro* and *Methanosarcina mazei Go1* are both of the genus Methanosarcina. Additionally, *Methanobacterium formicicum* and uncultured Methanobacteriales archaeon are of the order Methanobacteriales. These four microbes are collectively under the Euryarchaeota phylum, a collection of known methanogens. Methanogens are species that produce methane as a product of many of their metabolic processes, in addition to several of the common microbial metabolites introduced above [21]. This methane-producing behavior is rather unique to archaeal species. In analyzing breath samples, methane gas-to-hydrogen gas ratio has been shown to differ significantly between cancerous patients and healthy normal, even in studying non-lung cancers [22]. Microbial fermentation, as practiced by these methanogenic archaeal species, is thought to account for the pronounced methane presence among cancer samples. It is possible that synthesis of methanogenic compounds directly within lung tissue might *contribute* to the development of cancer, as observed in patients with higher breath methane-content. Heavy DNA methylation is known to exhibit cancerous effect through a variety of mechanisms [23,24]; one commonly observed example involves extensive methylation of tumor suppressor genes, preventing promotor recognition, transcription, and subsequent translation of necessary cellular proteins [23,24]. It is plausible that intra-tissue production of this compound may contribute to the methylation process. When produced at sites more proximal to cells, especially if at considerable concentration, methane and the many other similar metabolites of these archaea might contribute to increased methylation of an individual’s genetic material. This would serve to increase the likelihood of cancer development and progression.

*Thermoproteus* sp. *IC-062* and *Pyrodictium brockii* are of the class Thermoprotei. In particular, Thermoproteus species reduce elemental sulfur as a critical step in their metabolic processes. More commonly, they are also known to be capable of reduction in polysulfides and sulfates, generating thiosulfate and sulfide [25]. As described above, hydrogen sulfide is thought to contribute to cancer development through regulation of apoptotic and extracellular degradation pathways [7,20,26]. However, many of these sulfurous aspects are also present throughout cellular macromolecules: namely proteins. Though the effect of Thermoprotei interaction within human tissue has seen little investigation, several plausible mechanisms of influence can be deduced. Reduction in critical sulfurous components in human enzymes might dysregulate cellular function within a tissue sample. Disruption of enzymes associated with tumor suppression or enhancement of enzymes with oncogenic properties might both serve as mechanisms of impact from these species on human tissue. As such, this might also prove a logical means of correlation between these species’ abundances and the observed differences in cancer development.

Regarding investigation purely across cancer samples, 17 of the 37 differentially abundant LUAD species are also of the methane producing phylum Euryarchaeota. A total of 10 of the 18 significant species present across LUSC samples are of this phylum. Increase or decrease in metabolic effect of these species, as proposed above, might also serve to account for noted differences in cancer stage, neoplasm development, and effectiveness of therapy treatments.

Likewise, 13 significant species are of the class Thermoprotei among LUAD samples, and 5 are of the class across LUSC patients. The metabolic effects of these microbes may too be responsible to the discrepancies in cancer progression across patients as observed.

6 microbes significantly correlated to patient survival were also found to be differentially abundant across cancer samples: *Desulfurococcus amylolyticus 1221n*, *Desulfurococcus amylolyticus DSM 16532*, Methanothermobacter thermautotrophicus, *Pyrobaculum islandicum*, *Thermoproteus uzoniensis*, and uncultured Nitrosopumilales archaeon. These microbes’ abundance values correspond significantly to both patient survival length *and* patient cancer stage, therapy effectiveness, or neoplasm development. Hence, it might be understood that these species impact cancer progression to a considerable extent if over- or under-abundant in tissue. Again, their taxonomic categorization in the Euryarchaeota phylum or Thermoprotei class may provide a potential means of causation for the noted discrepancies in progression.

### 3.2. Molecular Targets

Interestingly, several methanogenic species have been found present in an array of food types [27]. Discovered in meat, vegetables, cheese, and most notably fish, these taxa likely enter the human body through consumption [27,28]. Methanogens are only present in relatively small quantities; however, this is consistent with the abundance levels as estimated above within lung tissue samples. Potential migration of these microbes from the gastrointestinal tract to other bodily systems, particularly the lungs, would account for the measured archaeal counts in lung tissue. As such, dietary regulation may be capable of reducing bodily presence of particular archaeal species, ultimately decreasing an individual’s risk of developing lung cancer. Indeed, dietary alteration is known to influence microbiome composition [29]. Frequent consumption of many fermented carbohydrates is known to sustain species of several common genera, including Bifidobacterium, Prevotella, Ruminococcus, Dorea, and Roseburia; high-fat diets have been shown to increase abundance of bile-resistant organisms, such as Bilophila and Bacteroides [29]. Malnutrition has even seen certain correlation to microbiome change and composition [30]. Regarding the archaeal domain, several species of the Thermoproteota (Crenarchaea) phylum are known to exist in many fermented seafoods [31]; limiting consumption of these food may therefore reduce abundance of key archaeal species, ultimately minimizing the likelihood of lung cancer development or progression.

Alternatively, rather novel means of microbiome composition alteration are under development. Targeted radiotherapy treatments have been shown capable of direct species abundance manipulation, even of tumoral microbiomes [32,32]. Specific species are known to exhibit varying responses when exposed to radiation of particular frequency. In limiting the growth and replication of unwanted species within tumoral tissue, a cancer’s likelihood of progression can be significantly decreased [32,33]. Microbial radiotherapy may enable tumoral microbiome manipulation, allowing a direct means of regulating abundance levels of select archaeal species.

Ultimately, attempt to regulate archaeal presence proves difficult due to lack of research; little investigation has attempted to characterize the means by which these species are introduced to the body. Logically, targeting their metabolic products might serve as a more reasonable means of limiting the risk of cancer development and progression. Indeed, many natural products are known to exhibit carcinogenic effect, capable of altering genetic material and cellular pathway regulation [34]. These products are broadly classified into two categories: genotoxic agents which cause direct gene damage, and non-genotoxic agents which encourage carcinogenesis by some secondary means. Whether genotoxic or non-genotoxic, many microbial metabolites are known to be carcinogenic [5,6,7]. Reactive oxygen species (ROS), as well as methane, sulfur, and their chemical derivatives, may be appropriate metabolic targets in attempt to hinder oncogenesis. In fact, there exist several known means of regulating these metabolites at a cellular level.

ROS are highly reactive compounds, capable of oxidizing vital cellular proteins and macromolecules [5,6]. Nonetheless, oxidative stress caused by ROS is reduced through regular intake of antioxidants (e.g., beta-carotene, vitamin B) [35]. These molecules react directly with ROS, significantly decreasing the likelihood of alternate reactions with necessary cellular molecules [5,6]. In particular, regular consumption of vitamin B6 and vitamin B12 has been shown to increase oxidative metabolism, leading to decreased levels of oxidative stress and inflammation [35]. A decrease in presence of cellular oxidative compounds reduces opportunity for radical reaction and potential carcinogenesis [5,6]. In addition to synthetic antioxidants, many fruits and vegetables are rich in antioxidant vitamins [36]. Maintaining an adequate source of these nutrients has been proven to limit the risk of cancer development, with relation to regulation of oxidative stress and inflammation [35].

In this way ROS reaction with methane- and sulfur-derived species would also see a decrease. ROS chemically activate molecules of which they react with [28]; hence, with lesser ROS presence, methane and sulfurous compounds would be less capable of further activation. This might reduce their likelihood of undesirable reaction with cellular components, effectively reducing the risk of cancer development and progression.

Irrespectively, GSEA revealed there is in fact a correlation between the archaeal species differentially abundant in LUAD and LUSC tissue and the expression of select genes in lung cells. Several known oncogenic cellular signal pathways were either positively or negatively enriched with respect to these genes; this implies there also exists a correlation between the significant taxa and cellular oncogenic tendency. Further investigation into the intricacies of these relations might reveal a specific influence of archaeal species on oncogenic and tumor suppressive gene expression. This would ultimately detail a more-concise mechanism by which archaeal species might contribute to cancer development or progression.

GSEA also revealed a metabolomic correlation to archaeal abundance in cancer samples. With several cellular pathways significantly dysregulated, these significant archaeal species have the molecular ability to influence cellular metabolism. The discovered correlation of archaeal abundance to metabolic gene expression may act as a mechanism for archaeal-associated differences in cancer incidence and progression. Indeed, cancer cells are known to contain considerable alteration in metabolic pathway regulation [37]; this can be largely attributed to increased demand of energy with continued proliferation. If archaeal species are also capable of upregulating key metabolic pathways necessary for tumorigenesis, this may explain the measured association between archaeal abundance and oncogenesis. Select archaeal species, as identified above, afford cells additional metabolic means of unregulated growth and replication. Perhaps in targeting these specific metabolic pathways, archaeal influence on carcinogenesis can be mitigated.

As demonstrated with the predictive machine learning-based diagnostic algorithms, the archaeal microbiome can be used as a *highly* distinctive indicator of both LUAD and LUSC incidence among patients. Particular abundance values of select features allows for prediction of cancer presence with over 99% accuracy among both cancer types. This means of diagnosis is limited in the fact that *tumor tissue* must be extracted for comparison, meaning cancer incidence is already known. Recent studies, however, have seen success in cancer diagnosis with use of mere blood samples. With microbial DNA also present in blood plasma, variances in microbiome were used to train a machine learning-based algorithm to diagnose 33 types of cancer with great accuracy. Perhaps inclusion of archaeal species in models like these would increase prediction accuracy and robustness.

Nonetheless, the implications of the distinct archaeal microbiome of LUAD and LUSC tumor tissue samples are significant. With unique metabolic properties, archaeal species present in lung tissue may be of causational correlation to cancer development and progression. Many produced metabolites are known to promote carcinogenesis both through direct genomic damage and cellular signaling pathway regulation, In the event of these occurrences, archaeal-induced tumorigenesis is plausible. Further investigation into the intricacies of specific metabolite-host interactions may be useful in elucidating the microbiome’s role in lung cancer incidence. Moreover, research regarding the microbiome’s influence on cancer development and therapy efficacy may see clinical applicability, allowing additional considerations in cancer staging, survival estimation, and creation of optimized treatment plans. Ultimately, the archaeal microbiome must be further explored to understand its influence on human physiological function, immunity, and disease.

## 4. Methods

### 4.1. TCGA Data Acquisition

Raw whole-transcriptome RNA-sequencing data was downloaded from the TCGA legacy archive for 527 LUAD samples, 479 LUSC samples, and 99 normal adjacent lung tissue samples (https://portal.gdc.cancer.gov/legacy-archive/search/f) (accessed on 1 August 2021). Corresponding clinical data of these patients was downloaded from the Broad GDAC Firehose (https://gdac.broadinstitute.org/, accessed on 1 August 2021).

### 4.2. Microbial Abundance Extraction and Computation

RNA-sequencing data was filtered for microbial abundance counts with the Pathoscope 2.0 program using an archaeal sequence library assembled from the NCBI nucleotide database (https://www.ncbi.nlm.nih.gov/nucleotide/, accessed on 10 September 2021). The Pathoscope 2.0 program generates two distinct abundance approximations: a “best guess” estimating the relative abundance of each species, as a percentage, and a “best hit” estimating the exact amount of each species, expressed as an integer. Best hit values were used throughout the above analyses.

### 4.3. Microbial Contamination Correction

Spearmen’s rank correlation coefficients were used to identify likely contaminant microbes. This means of correlation describes the monotonicity of relation between two variables: individual species abundance and total microbe abundance. A threshold value of *p* < 0.05 was used to distinguish significant microbes from insignificant microbes.

### 4.4. MDS Ordination Application

Multidimensional Scaling (MDS) ordination was used to characterize the microbial landscape of each LUAD, LUSC and normal sample. MDS ordination reduces microbial abundance values of each species within a sample into two dimensions, allowing for simpler visualization of differences across samples. Bray–Curtis ordination was employed, in which samples serve as the entities, and species abundance values serve as the attributes. Computationally, a dimensional matrix was calculated for each cohort, axis direction was determined, and all appropriate samples were plotted within each matrix according to their microbial abundance reads.

### 4.5. β-Diversity Assessment

β-diversity describes the extent of variation across samples of a defined origin. Species often present in samples of one cancer status, but not others, would increase the amount of diversity between groups. Likewise, species common to both cancer status groupings of concern would decrease overall diversity. As such, Β-diversity scores were calculated across all samples within a cancer status grouping for comparison.

### 4.6. Differential Abundance Determination

The cancer statuses of patients were aligned with microbial read counts of each species. Clinical values were also aligned for each of the seven variables of concern. Kruskal–Wallis testing was conducted, revealing microbes of significant relation to cancer status and each clinical variable, with *p* < 0.05 used as a threshold value. Corresponding box plots were created, each displaying median and upper- and lower- quartile abundances of each species within each variable.

Microbe counts were simplified to a binary “high” or “low” value, depending on their relation to the median microbe count. Patients’ classifications for each differentially abundant species were aligned with their respective gene expression counts. Kruskal–Wallis testing revealed genes of significantly altered expression in relation to “high” or “low” species presence.

### 4.7. Survival Discrepancy Determination

Patient alive statuses and times of assessment were aligned with microbial counts for survival analyses. Survival analyses were conducted using Cox Regression, revealing microbes of which higher or lower abundance corresponded to increased or decreased patient survival; microbe counts of patients were compared across patients of differing alive statuses throughout an assessed period of time, with *p* < 0.05 used as a threshold value. Microbe counts were simplified to a binary “high” or “low” value, depending on their relation to the median microbe count in order to increase concision of conclusions. Corresponding survival plots were created, each displaying the proportions of “high” abundance and “low” abundance patients through a given timeframe.

### 4.8. Supervised Machine Learning Classification

The LGBM machine learning algorithm was chosen to classify LUAD and LUSC patients from normal samples, consistently outperforming similar machine learning algoritms [38,39]. The classifier was implemented using official python packages from Microsoft. As the dataset is unbalanced (more cancer samples than normal adjacents), the weight of dominate cancer labels were proportioned according to their fractions. Data was randomly splits into 70% training cohorts and 30% testing cohorts, then normalized using only training data. RFE was implemented using sklearn, with 5-fold cross validation and default parameters. SHAP value was computed using the original package provided [40].

### 4.9. Gene Set Enrichment Analysis (GSEA)

189 oncogenic gene sets and 225 metabolomic gene sets were sourced from the Molecular Signaling Database’s (MSigDB) hallmark gene set collection [41]. Read counts of LUAD and LUSC differentially expressed genes, as derived through Kruskal–Wallis testing above, were analyzed using GSEA software [42,43]. Cancer status served as the phenotype of concern.

## Figures and Tables

**Figure 1 ijms-23-11566-f001:**
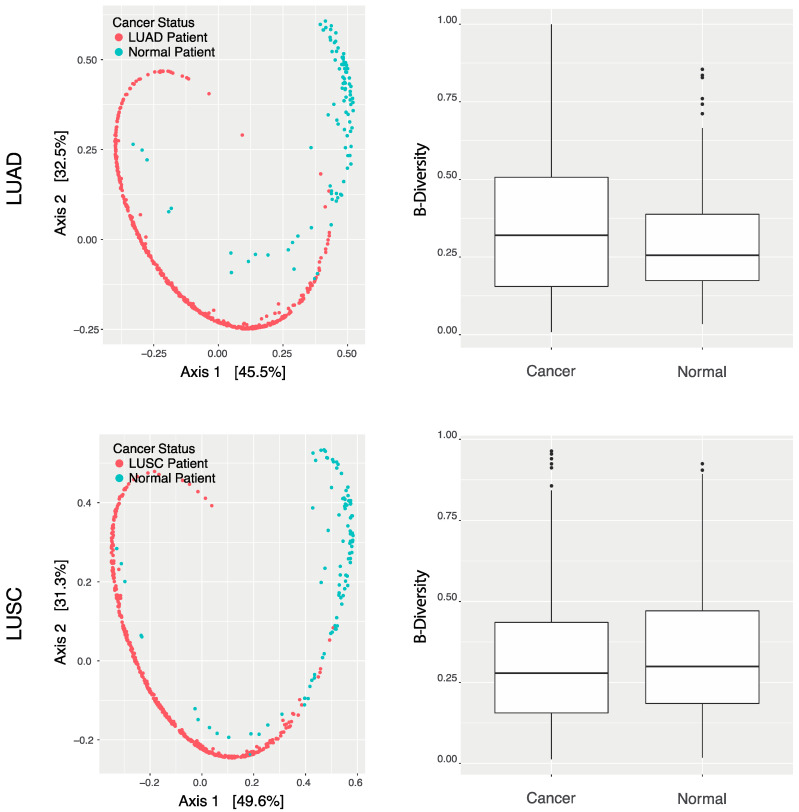
Wholistic lung sample microbiome analysis. Multidimensional Scaling Bray–Curtis Ordination plots of LUAD and LUSC (red), and normal patient (turquoise) microbial landscapes. Bar chart of β-diversities of all mapped species by LUAD, LUSC, and normal samples.

**Figure 2 ijms-23-11566-f002:**
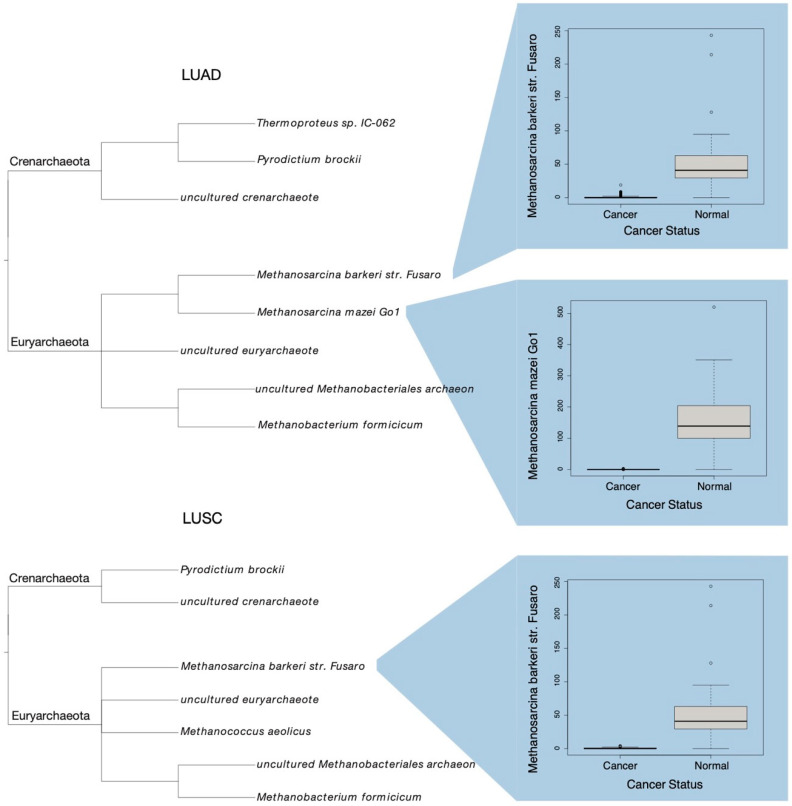
Phylograms of species differentially abundant in LUAD and LUSC samples, as compared to normal adjacent samples. Respective taxonomic phylums are shown. Select box plots of species abundance in cancer samples versus normal samples.

**Figure 3 ijms-23-11566-f003:**
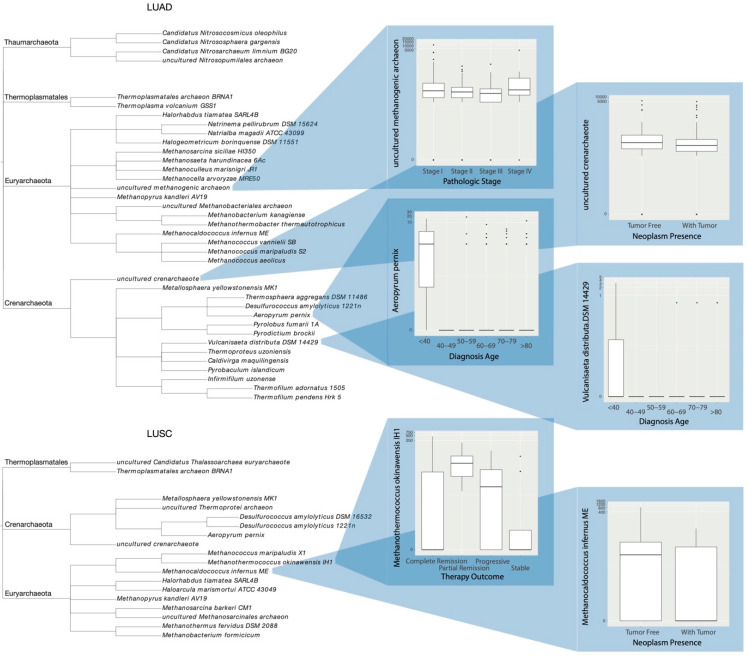
Phylograms of species differentially abundant among LUAD and LUSC samples, with respect to patient clinical categorization. Respective taxonomic phylums are shown. Select box plots of species abundance by samples of varying clinical status.

**Figure 4 ijms-23-11566-f004:**
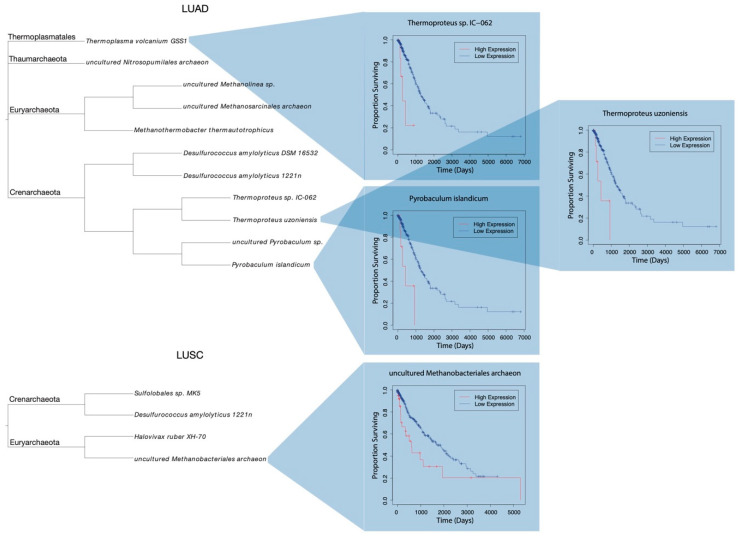
Phylograms of species differentially abundant among LUAD and LUSC samples, with respect to patient clinical categorization. Respective taxonomic phylums are shown. Patients were classified as “high” or “low” abundance of a species. Select Kaplan–Meier survival plots of proportion of “high” and “low” abundance patients surviving with time.

**Figure 5 ijms-23-11566-f005:**
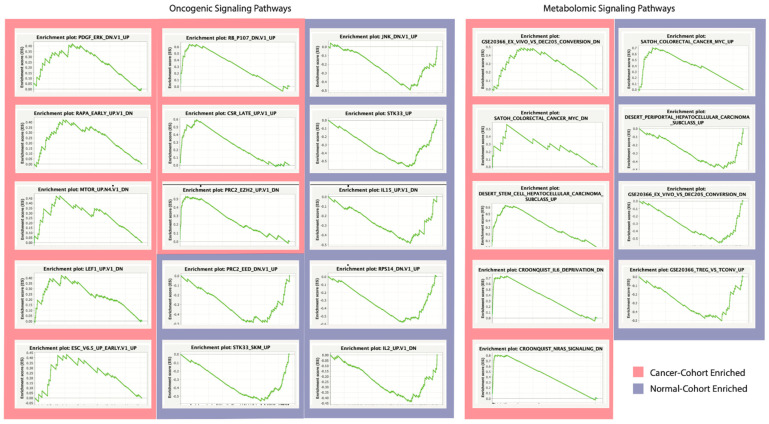
GSEA Enrichment Plots. Gene sets significantly enriched in cancer sample cohorts (red) or normal sample cohorts (blue), with enrichment score magnitude and direction. Division of oncogenic associated and metabolomic associated signaling pathways of enrichment dysregulation. Signaling pathways of the largest peak height exhibit the greatest extent of dysregulation.

**Figure 6 ijms-23-11566-f006:**
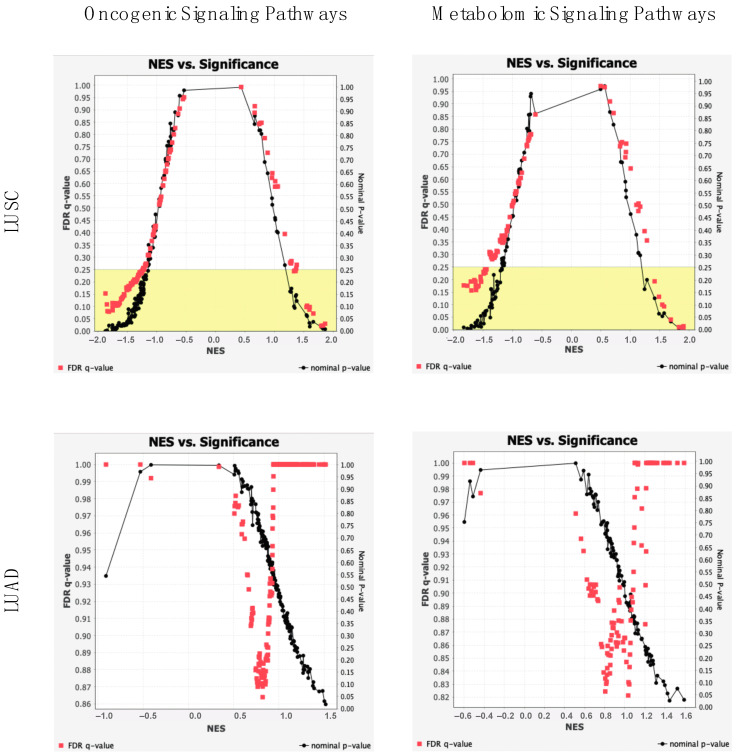
GSEA Significance Plots. Displays of pathway NESs with associated nominal p-values (black) and FDR q-values (red) for LUAD and LUSC analyses. Division of oncogenic associated and metabolomic associated signaling pathways. Signaling pathways of the left and right extremes of graphs exhibit the greatest extent of dysregulation, while signaling pathways of the bottom portion are of the greatest significance. Yellow shaded region signifies q-values below 0.25.

**Figure 7 ijms-23-11566-f007:**
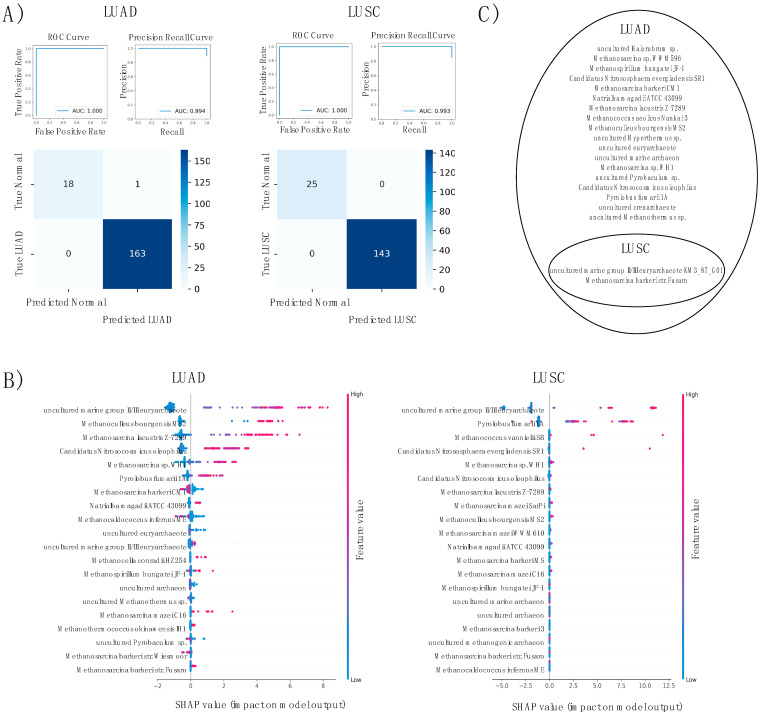
Machine learning classification of LUAD and LUSC. (**A**) ROC, precision recall curve and confusion matrix. Classifiers of the largest AUC, with values between 0 and 1, exhibit the greatest predictive accuracy. (**B**) SHAP values breakdown of two trained models. Features of the largest SHAP values are of greatest influence to the models’ predictive means. (**C**) Remaining features after RFE by cancer type. Most significant species to model prediction succeeding elimination.

**Table 1 ijms-23-11566-t001:** List of microbes differentially abundant across LUAD and normal samples and LUSC and normal samples. Corresponding *p*-Value denoting significance of relationship.

Microbe	LUAD vs. Normal	LUSC vs. Normal
*Methanobacterium formicicum*	1.37 × 10^−15^	3.88 × 10^−19^
*Methanococcus aeolicus*		0.035682
*Methanosarcina barkeri str. Fusaro*	3.45 × 10^−45^	1.42 × 10^−49^
*Methanosarcina mazei Go1*	1.70 × 10^−101^	
*Pyrodictium brockii*	0.026487	0.014115
*Thermoproteus* sp. *IC-062*	0.011968	
*uncultured crenarchaeote*	1.09 × 10^−35^	1.32 × 10^−36^
*uncultured euryarchaeote*	1.12 × 10^−50^	4.92 × 10^−52^
*uncultured Methanobacteriales archaeon*	0.000271	0.004886

**Table 2 ijms-23-11566-t002:** List of microbes differentially abundant across LUAD and LUSC samples. Corresponding *p*-Value denoting significance of relationship.

Cancer	Microbe	Diagnosis Age	Sex	Race	Smoking History	Pathologic Stage	Neoplasm Presence	Therapy Outcome
LUAD	*Aeropyrum pernix*	0.000339				0.027172		
	*Caldivirga maquilingensis*					0.01464		
	*Candidatus Nitrosarchaeum limnium BG20*	0.012235			0.011974			
	*Candidatus Nitrosocosmicus oleophilus*	0.000243	0.026028					
	*Candidatus Nitrososphaera gargensis*					0.026028		
	*Desulfurococcus amylolyticus 1221n*					0.034768		
	*Halogeometricum borinquense DSM 11551*		0.015319					
	*Halorhabdus tiamatea SARL4B*	0.047583						
	*Infirmifilum uzonense*				0.003069			
	*Metallosphaera yellowstonensis MK1*				0.036199			
	*Methanobacterium kanagiense*			0.029399				
	*Methanocaldococcus infernus ME*		0.022004					
	*Methanocella arvoryzae MRE50*	0.047327						
	*Methanococcus aeolicus*				0.002794			
	*Methanococcus maripaludis S2*	0.001452						
	*Methanococcus vannielii SB*	0.001327						
	*Methanoculleus marisnigri JR1*	1.42 ×10^−8^		0.000996				
	*Methanopyrus kandleri AV19*		0.024732		0.007884			
	*Methanosaeta harundinacea 6Ac*	0.000167						
	*Methanosarcina siciliae HI350*			0.041515				
	*Methanothermobacter thermautotrophicus*	0.001041						
	*Natrialba magadii ATCC 43099*	0.022468						
	*Natrinema pellirubrum DSM 15624*	8.59 × 10^−12^						
	*Pyrobaculum islandicum*					0.016035	0.046881	0.000782
	*Pyrodictium brockii*			0.044358				
	*Pyrolobus fumarii 1A*			0.037196		0.025456		
	*Thermofilum adornatus 1505*				0.037196			
	*Thermofilum pendens Hrk 5*	0.000006						
	*Thermoplasma volcanium GSS1*	0.042784						
	*Thermoplasmatales archaeon BRNA1*				0.02283			
	*Thermoproteus uzoniensis*					0.016035	0.046881	0.000782
	*Thermosphaera aggregans DSM 11486*	5.79 × 10^−7^						
	*uncultured crenarchaeote*					0.045191	0.010931	
	*uncultured Methanobacteriales archaeon*	0.007569						
	*uncultured methanogenic archaeon*			0.002368	0.009116			
	*uncultured Nitrosopumilales archaeon*	0.00077						0.032531
	*Vulcanisaeta distributa DSM 14429*	0.000003						
LUSC	*Aeropyrum pernix*						0.030009	
	*Desulfurococcus amylolyticus 1221n*	0.006251						
	*Desulfurococcus amylolyticus DSM 16532*	0.002899						
	*Haloarcula sp CBA1115*							0.00029
	*Halorhabdus tiamatea SARL4B*				0.005232			
	*Metallosphaera yellowstonensis MK1*	0.032987						
	*Methanobacterium formicicum*							0.042146
	*Methanocaldococcus infernus ME*						0.007339	
	*Methanococcus maripaludis X1*	7.02 × 10^−24^						
	*Methanopyrus kandleri AV19*	0.018165						
	*Methanosarcina barkeri CM1*	0.013269					0.011716	
	*Methanothermococcus okinawensis IH1*							0.024171
	*Methanothermus fervidus DSM 2088*	0.007446				0.034249		
	*Thermoplasmatales archaeon BRNA1*					0.014448		
	*uncultured Candidatus Thalassoarchaea*					0.015503		
	*euryarchaeote*							
	*uncultured crenarchaeote*	0.02786						
	*uncultured Methanosarcinales archaeon*			0.001505				
	*uncultured Thermoprotei archaeon*				0.047705			

**Table 3 ijms-23-11566-t003:** List of survival-significant microbes across LUAD and LUSC samples. Corresponding *p*-Value denoting significance of relationship.

Microbe	LUAD	LUSC
*Desulfurococcus amylolyticus 1221n*	0.030359	0.049114
*Desulfurococcus amylolyticus DSM 16532*	0.034705	
*Halovivax ruber XH-70*		0.043801
*Methanothermobacter thermautotrophicus*	0.035661	
*Pyrobaculum islandicum*	0.002939	
*Sulfolobales* sp. *MK5*		0.026319
*Thermoplasma volcanium*	0.045003	
*Thermoproteus* sp. *IC-062*	0.002299	
*Thermoproteus uzoniensis*	0.002939	
*uncultured Methanolinea* sp.	0.010941	
*uncultured Methanosarcinales archaeon*	0.028039	0.958966
*uncultured Nitrosopumilales archaeon*	0.014496	
*uncultured Pyrobaculum* sp.	0.040101	

## Data Availability

All TCGA data can be accessed online through the TCGA data portal.

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
