# Peer review of "Archaea Microbiome Dysregulated Genes and Pathways as Molecular Targets for Lung Adenocarcinoma and Squamous Cell Carcinoma"

_ijms, 2022, doi:10.3390/ijms231911566_

Round 1

Reviewer 1 Report

 Uzelac et al demonstrated that the archaeal microbiome could be used as a highly distinctive indicator of both LUAD and LUSC incidence among patients.

Comment 1- The authors should explain the mechanisms associated with microbiome-induced lung carcinoma introduction section.

Comment 2- The authors should mention different factors involved in cancer pathogenesis and briefly highlight the role of inflammation and oxidative stress induced by gut microbiome in lung adenocarcinoma.

Comment 3- The link between different sentences and paragraphs is lacking. The authors are advised to recheck the introduction and provide some links.

Comment 4- The objectives of the study should be clear and well-written.

Comment 5- The importance of dietary factors in the modulation of microbiome and cancer progression should be more defined.

Comment 6- The importance of natural products in the modulation of cancer should be stated in the discussion.

Natural Products: Implication in Cancer Prevention and Treatment through Modulating Various Biological Activities. http://doi.org/10.2174/1871520620666200705220307.

Comment 7- The grammar should be checked.

Comment 8- Include a graphical summary.

Comment 9- Include the strategies to identify and differentiate different lung microbial compositions belonging to LUAD or LUSC from their normal counterparts.

Comment 10- Very limited and very basic information about the role of the microbiome is provided in the introduction. This could have been quite comprehensive.

Comment 11- Include the significance of the horseshoe effect to discover microbial niches across tumor and healthy environments.

Comment 12- Rewrite the conclusion part.         

Comment 13- The legends of the figures are very simple, they should be more informative.

Reviewer 2 Report

The Archaea microbiome in two lung carcinoma types was studied and compared with healthy samples. Interesting differences were described, which can pave the way for new lung cancer treatments. The manuscript is well written and accurately structured. I recommend acceptance after minor revision:

The presentation of some figures can be improved, some figures are too crowded. Maybe parts of the figures can be put into a supplement.

Discussion: Please discuss how far approved anticancer treatments can influence the lung microbiome and/or other human microbiomes. Some dietary components and vitamins were mentioned as antioxidants. Which other compounds appear to be promising? Specify or explain vitamin B in more detail.

The abbreviation ´´ROSs´´ is quite uncommon. Why not ´´ROS´´? Please explain.

The references 32-35 are presented in a different style. Please make conform according to the journal guidelines.
